# Heart Failure with Preserved Ejection Fraction and Pulmonary Hypertension: Focus on Phosphodiesterase Inhibitors

**DOI:** 10.3390/ph15081024

**Published:** 2022-08-19

**Authors:** Artem Ovchinnikov, Alexandra Potekhina, Evgeny Belyavskiy, Fail Ageev

**Affiliations:** 1Out-Patient Department, Institute of Clinical Cardiology, National Medical Research Center of Cardiology Named after Academician E.I. Chazov, 3-d Cherepkovskaya St., 15a, 121552 Moscow, Russia; 2Department of Clinical Functional Diagnostics, A.I. Yevdokimov Moscow State University of Medicine and Dentistry, Delegatskaya St., 20, p. 1, 127473 Moscow, Russia; 3Department of Internal Medicine and Cardiology, Campus Virchow Klinikum, Charité Universitätsmedizin Berlin, Augustenburger Platz, 13353 Berlin, Germany

**Keywords:** phosphodiesterase, PDE inhibitors, pulmonary hypertension, heart failure with preserved ejection fraction, diastolic dysfunction

## Abstract

Pulmonary hypertension (PH) is common in patients with heart failure with preserved ejection fraction (HFpEF). A chronic increase in mean left atrial pressure leads to passive remodeling in pulmonary veins and capillaries and modest PH (isolated postcapillary PH, Ipc-PH) and is not associated with significant right ventricular dysfunction. In approximately 20% of patients with HFpEF, “precapillary” alterations of pulmonary vasculature occur with the development of the combined pre- and post-capillary PH (Cpc-PH), pertaining to a poor prognosis. Current data indicate that pulmonary vasculopathy may be at least partially reversible and thus serves as a therapeutic target in HFpEF. Pulmonary vascular targeted therapies, including phosphodiesterase (PDE) inhibitors, may have a valuable role in the management of patients with PH-HFpEF. In studies of Cpc-PH and HFpEF, PDE type 5 inhibitors were effective in long-term follow-up, decreasing pulmonary artery pressure and improving RV contractility, whereas studies of Ipc-PH did not show any benefit. Randomized trials are essential to elucidate the actual value of PDE inhibition in selected patients with PH-HFpEF, especially in those with invasively confirmed Cpc-PH who are most likely to benefit from such treatment.

## 1. Introduction

Approximately half of patients with heart failure (HF) have preserved left ventricular ejection fraction (HFpEF) [1]. HFpEF is associated with substantial morbidity and mortality, and its prevalence increases due to the aging of the population and high incidence of arterial hypertension, obesity, and diabetes [2]. To date, no specific therapeutic agent except for sodium–glucose cotransporter-2 inhibitor empagliflozin [3] has been demonstrated to improve outcomes in patients with HFpEF. Moreover, most pharmacological agents have failed to improve the exercise capacity and quality of life in these patients [2].

The negative prognostic impact of HFpEF is predominantly associated with pulmonary hypertension (PH). PH is defined as a mean pulmonary artery pressure >20 mm Hg at rest estimated by cardiac catheterization [4]. In patients with HFpEF, pulmonary hypertension is associated with a poor prognosis [5,6]. The prevalence of PH in HFpEF ranges from 31% to 88%, depending on the severity of HFpEF, diagnostic criteria, and methods of evaluation (echocardiography or cardiac catheterization) [7,8,9,10,11,12,13]. The prevalence of PH in HFpEF is a result of a chronic increase in mean left atrial (LA) pressure, which is an important hemodynamic event in the natural course of HFpEF. Mean LA pressure is one of the three key determinants of mean pulmonary artery pressure along with right ventricular (RV) stroke volume and pulmonary vascular resistance. In the systemic circulation, the pressure in the underlying chamber (right atrium) is significantly lower than the systemic arterial pressure and does not play a significant role in maintaining the latter. On the contrary, in the pulmonary vascular bed, the pressure in the underlying chamber (LA) is an important component of mean pulmonary artery pressure, providing approximately half of its value. An increase in LA pressure is accompanied by a corresponding rise in mean pulmonary artery pressure. In the early stages of HFpEF, mean LA pressure and mean pulmonary artery pressure only increase with exercise. Later, the mean LA pressure becomes chronically elevated at rest, which is manifested by an increase in pulmonary capillary wedge pressure >15 mm Hg, and leads to a proportional rise in mean pulmonary artery pressure (isolated postcapillary PH, Ipc-PH) [14,15]. The transpulmonary pressure gradient (a difference between mean pulmonary artery pressure and pulmonary capillary wedge pressure) and diastolic pulmonary gradient (a difference between diastolic pulmonary artery pressure and pulmonary capillary wedge pressure) are maintained within ≤12 mm Hg and <7 mm Hg, respectively [16]. In Ipc-PH, structural changes are localized in the pulmonary venule and capillaries, while the pulmonary arteries remain intact, and the pulmonary vascular resistance is below 3 Wood units (a simplified system for measuring pulmonary vascular resistance; 1 Wood unit = 80 dyns × sec × cm^−5^ according to the formula: pulmonary vascular resistance = (mean pulmonary arterial pressure − pulmonary capillary wedge pressure)/cardiac output) [17].

With a prolonged increase in mean LA pressure, the reactive “precapillary” alterations of pulmonary vasculature complicate the passive “postcapillary” component of PH. These reactive changes initially occur in small pulmonary arteries and arterioles as functional (vasoconstriction) and then as structural alterations (structural remodeling) with the development of the combined pre- and post-capillary PH (Cpc-PH) [18,19]. Structural remodeling is triggered by a cascade of proinflammatory and profibrotic reactions and is referred to as pulmonary vascular disease. The development of pulmonary vascular disease leads to an increased resistance to blood flow in the pulmonary circulation, which is manifested as an increase in pulmonary vascular resistance and a disproportionate elevation of mean pulmonary artery pressure compared to the mean LA pressure. Such changes are represented as an increase in pulmonary vascular resistance >3 Wood units, transpulmonary pressure gradient >12 mm Hg, and/or diastolic pulmonary gradient ≥ 7 mm Hg (Figure 1).

## 2. Mechanisms of the Combined Pre- and Post-Capillary PH

The pathogenesis of Cpc-PH in HFpEF and the mechanisms of activation of reactive pulmonary changes have not been fully elucidated. Only a portion of patients with chronically elevated mean LA pressure develop the reactive vascular changes [20]. On the other hand, some patients with early HFpEF and normal resting LA pressure demonstrate pulmonary vascular dysfunction and blunted vasodilatory pulmonary reserve [21,22]. Patients with HFpEF and Cpc-PH have similar risk factors and comorbidities to those with Ipc-PH [9,20]. However, patients with Cpc-PH were demonstrated to be younger than patients with Ipc-PH [23]. The causes of pulmonary vascular remodeling in HFpEF are not limited to hemodynamic factors, but also include pro-inflammatory metabolic pathways [24].

Possible pathophysiological mechanisms of structural pulmonary changes and pulmonary hemodynamic derangements include decreased nitric oxide production and increased endothelin-1 production in pulmonary vascular endothelium, decreased soluble guanylate cyclase (GC) activity, and activation of adrenomedullin neurohumoral pathways [21,25,26,27]. In addition, hypoxia impairs electrolyte and intra-alveolar fluid transport [28], inducing vasoconstriction and the structural remodeling of small pulmonary arteries [29].

A prolonged increase in mean LA pressure (>20–25 mm Hg) is obligatory for the development of pulmonary vascular disease [30], and depends on the severity of underlying left ventricular (LV) diastolic dysfunction and existing LA myopathy. Preserved LA function prevents an increase in pulmonary venous pressure despite existing LV diastolic dysfunction, and effectively protects the pulmonary microvasculature and right heart chambers from the adverse effects of increased LV filling pressure [31]. Conversely, LA dysfunction leads to vulnerability to the damaging effects of high LV filling pressure and the development of PH [31,32] and RV failure [31,33]. Since an increase in mean pulmonary artery pressure is characteristic of all patients with elevated LA pressure, left-sided PH can be considered a separate stage in the natural course of HFpEF.

## 3. Epidemiology of the Combined Pre- and Post-Capillary PH in HFpEF

Ipc-PH is the most prevalent form of PH in patients with HFpEF [7]. Since cardiac catheterization in HFpEF is not performed routinely, the accurate estimation of the incidence of the increase in pulmonary vascular resistance is challenging. Gergers et al. identified diastolic pulmonary gradient ≥ 7 mm Hg (an invasive marker of Cpc-PH) in 23% of patients with HFpEF and PH among nearly 4000 cardiac catheterizations [9]. Among 2587 patients with HFpEF and PH, a diastolic pulmonary gradient ≥ 7 mm Hg was detected in 12% of patients, pulmonary vascular resistance ≥ 3 Wood units in 36% of patients, and transpulmonary pressure gradient > 12 mm Hg in 46% of patients [6]. The identification of pulmonary vasculopathy in PH-HFpEF is critical, not only because patients with Cpc-PH have a poorer prognosis, but also because they may benefit from specific treatment. Patients with Cpc-PH are anticipated to respond differently to pulmonary arterial hypertension (PAH)-specific therapy compared to patients with Ipc-PH [34,35].

## 4. Structural Changes in Pulmonary Vessels in PH-HFpEF

The response of pulmonary vessels to an increased LA pressure depends on the rate of its increase. In the case of an acute increase in LA pressure (e.g., due to paroxysmal atrial fibrillation or severe increase in blood pressure), the pulmonary capillaries are exposed to barotrauma. A further process, known as “alveolar–capillary stress insufficiency”, is manifested by an increase in the permeability of the endothelium and a violation of gas exchange. This is the mechanism for the development of acute pulmonary edema. In Ipc-PH, the mean pressure in the left atrium increases slowly and capillary barotrauma does not occur. The risk of pulmonary edema is minimal due to adaptive mechanisms reducing capillary permeability, including thickening of the alveolar–capillary membrane and enhanced lymphatic drainage (Figure 1) [36,37]. The main vascular alterations in the pulmonary veins are parietal thickness, deposition of collagen, and narrowing of the lumen—a process known as arteriolarization of the pulmonary veins [38,39]. The elimination of the Ipc-PH trigger (i.e., LA decompression) may help restore capillary and venous changes if the arteriolarization of the pulmonary veins has not occurred [38,40,41].

In Cpc-PH, structural remodeling initially affects pulmonary arterioles and small arteries, but further affect medium and large arteries, including pulmonary artery branches, resulting in decreased compliance of the pulmonary arteries [42,43]. The severity of PH has been shown to be closely associated with intimal thickening of pulmonary venules and arterioles [39]. At the Cpc-PH stage, structural changes in arterioles are manifested by the destruction of the inner elastic membrane, intimal and adventitial fibrosis, medial hypertrophy, and muscularization of arterioles, leading to luminal narrowing or occlusion (Figure 1) [44].

There is increasing evidence of an overlap between the advanced stages of PH-HFpEF and those of the PAH phenotype [45]. An exploratory genetic analysis in Cpc-PH identified genes and biological pathways in the lung known to contribute to PAH pathophysiology [23]. However, in contrast to pulmonary arterial hypertension, in Cpc-PH, pulmonary venules are actively involved in vascular remodeling [38,43,46].

Patients with Cpc-PH demonstrate low exercise tolerance, comparable to patients with PAH [47,48]. Poor exercise tolerance in patients with HFpEF and Cpc-PH is associated with blunting RV contractile reserve, which causes LV underfilling, insufficient cardiomyocyte stretching, and decreased LV contractility (according to the Frank–Starling mechanism) [49].

## 5. Right Ventricular Dysfunction in the Combined Pre- and Post-Capillary PH

RV dysfunction negatively affects the prognosis in left-sided PH. RV afterload increases in Cpc-PH due to elevated pulmonary vascular resistance (resistive load), while pulmonary vascular compliance decreases, which also increases the resistance to RV ejection (pulsatile load) [50]. In the systemic circulation, compliance is determined mainly by the proximal portion of the aorta. In the pulmonary vasculature, compliance is primarily determined by the distal vessels [51]. Elevated LA pressure increases the pulsatile load on the RV by reducing pulmonary vascular compliance, thus establishing the pathophysiological basis for the development of RV remodeling at the stage of “venous” PH, prior to the reactive changes [38,52]. Pulmonary and cardiac right-sided functional abnormalities (blunted RV and pulmonary vasodilatory reserves) are detected in patients with early stages of HFpEF, when LV filling pressures and pulmonary pressures are normal at rest and only increase during exercise [21,22].

The RV is poorly adapted to resist pressure overload [53], and despite adaptive responses (hypertrophy, increased contractility), most patients develop RV dysfunction and dilatation (Figure 1) [41,50]. The pulmonary vasculature is characterized by a high compliance, resulting in RV afterload ≈ 10 times lower than that of the LV [54]. For the pumping of stroke volume, the RV needs to exert an intracavitary pressure of only ≈30 mm Hg [54]. The RV is a thin-walled structure with limited capacity for adaptive hypertrophy. RV systolic pressure loading leads to high RV wall stress, which cannot be neutralized by the thin walls of the ventricle (according to Laplace’s law) and leads to a decrease in contractility [55]. The systolic dysfunction developed by the mechanism of afterload mismatch (pulmonary artery–RV uncoupling) can be almost completely reversible with a decrease in pulmonary vascular resistance.

Myocardial ischemia due to high RV systolic stress and systolic compression of the subendocardial coronary vessels (blood flow through the right coronary artery occurs predominantly in systole) also contributes to the decrease in RV contractility. High RV stress increases myocardial oxygen demand, exacerbating ischemia.

The main consequence of RV failure is an increase in central venous pressure, resulting in impaired lymphatic drainage from the lungs, excessive interstitial fluid accumulation, impaired gas exchange, reduced lung distensibility [56], and congestion in the systemic circulation. Liver and intestine congestion causes cholestasis, impaired absorption of nutrients and drugs, ascites, and the penetration of intestinal microflora into the bloodstream with the development of systemic proinflammatory status [57]. All of these processes lead to cachexia, which often accompanies RV dysfunction [58].

RV failure is associated with LV cardiac output reduction due to the blunting of the RV contractility reserve (serial ventricular interdependence) and LV underfilling, aggravating the symptoms of LV failure. With the RV dilatation and filling pressure increase, the parallel ventricular interdependence also increases, which is accompanied by a decrease in transmural LV diastolic pressure [59], shifting of the interventricular septum to the left, and impaired LV filling (Figure 1). Unloading the RV with diuretics or venous vasodilators (e.g., nitrates) often improves LV filling by restoring transmural diastolic pressure, although the LA pressure may decrease [60].

An increase in central venous pressure and a decrease in renal filtration pressure contribute to a reduction in sodium excretion and fluid retention, increase in circulating blood volume, and cardiorenal syndrome [61]. These processes induce blood congestion and further increase in both central venous pressure and LA pressure (Figure 1).

The key event in the natural history of PH in HFpEF is the “turning on” of the pulmonary reactive precapillary component, which increases RV afterload and RV dysfunction and worsens the prognosis [62,63], creating a vicious circle and accelerating the progression of HF (Figure 1). The only way to discontinue is to lower the pulmonary vascular resistance.

## 6. Pulmonary Hypertension Treatment in HFpEF

PH and RV dysfunction are associated with poor prognosis in HFpEF, and the effective treatment of PH is expected to extrapolate into a reduced risk of death. The method of reducing the mean pulmonary artery pressure depends on the type of left-sided PH. The goal of Ipc-PH treatment is to reduce the LA pressure, for example, by hemodynamic unloading of the heart. In clinical studies, LV assist devices have shown beneficial effects on PH in patients with terminal HF [64,65]. In patients with persistent PH after LV assist device implantation, the PDE5 inhibition with sildenafil was associated with a decrease in mean pulmonary artery pressure by ≈ 30% and in pulmonary vascular resistance by ≈ 50% [66]. Treatment with low-dose loop diuretics in euvolemic patients with HFpEF and PH was accompanied by a significant decrease in LV filling pressure and pulmonary artery systolic pressure [67].

The main target for Cpc-PH treatment is pulmonary vascular disease (increased pulmonary vascular resistance). Pulmonary vasodilators are highly effective in PAH [19]. The established overlap between the hemodynamic profile of Cpc-PH and PAH have initiated an active investigation of PAH-specific therapy in HFpEF [38], especially due to lack of effective treatment [2].

The effect of pulmonary vasodilators on pulmonary vascular resistance and mean pulmonary artery pressure depends on the severity of the structural changes in pulmonary vessels. Cpc-PH includes both pulmonary vascular remodeling and pulmonary vasoconstriction. Autopsy studies did not reveal a clear association between pulmonary vascular pathology and pulmonary vascular resistance, suggesting the involvement of functional vasoconstriction in the pathogenesis of Cpc-PH [68]. This is also evidenced by the acute decrease in pulmonary vascular resistance during pulmonary vasodilator treatment in patients with left-sided PH, demonstrating the role of a reversible component of active pulmonary vasoconstriction [69,70]. PAH-specific vasodilators restoring functional vasoconstriction have been demonstrated to prevent or regress the structural remodeling of the pulmonary vessels [71,72,73,74], and an effect on pulmonary vascular resistance in advanced Cpc-PH has been anticipated. Phosphodiesterase type 5 (PDE5) inhibitors have been shown to safely restore the pulmonary vascular changes in left-sided PH, thereby effectively unloading the RV [75].

## 7. Phosphodiesterases and Their Inhibitor: General Presentation

As is known, 3′5′-cyclic nucleotide phosphodiesterases (PDEs) belong to the superfamily of enzymes that hydrolyze two important secondary intracellular messengers: cyclic guanosine monophosphate (cGMP) and cyclic adenosine monophosphate (cAMP), which are involved in the regulation of the contraction and relaxation of smooth and cardiac myocytes. Nonetheless, there are many other families of phosphodiesterases, including phospholipases C and D, DNases and RNases, autotaxin, sphingomyelin phosphodiesterase, and restriction endonucleases, as well as numerous less well-characterized small-molecule phosphodiesterases. Seven cyclic nucleotide PDEs isoforms are expressed in the heart and only PDE5 and PDE9 are selective for cGMP [76,77].

cGMP is originated from guanosine triphosphate by the enzyme GC, persisting in two forms: soluble (cytosolic) and particulate (membrane). Soluble GC is a receptor for nitric oxide, and the particulate GC is a receptor for natriuretic peptides (Figure 2) [78,79]. cGMP stimulates protein kinase G. The following cascade includes the phosphorylation of enzymes of the intracellular calcium ion (Ca^2+^) cycle, channels, and structural proteins, leading to a decrease in smooth muscle tone, the suppression of β-adrenergic responses and pro-hypertrophic intracellular signals, a reduction in cardiomyocyte stiffness, antifibrotic effects, and the activation of pro-survival intracellular signaling pathways [76,80,81,82] (Figure 2).

cGMP also activates cGMP-binding proteins and cyclic nucleotide-gated channels [83]. The anchoring protein for protein kinase G is the same A-kinase anchoring protein (AKAP) as for protein kinase A, thus providing close colocalization and regulation of cAMP and cGMP activities and their intracellular signaling cascades [76]. cGMP controls cAMP levels by inversely modulating PDE2 and PDE3 [84]. At low concentrations, cGMP has a positive inotropic effect associated with an increase in cAMP, while at high concentrations, cGMP has an antiadrenergic effect and counteracts protein kinase A-mediated cardiac pro-hypertrophic signaling (Figure 2) [80].

The most well-studied member of the phosphodiesterase superfamily is PDE5. The structural assembly of PDE5 is a homodimer, containing both a regulatory GAF domain responsible for cGMP binding and a catalytic domain [85]. PDE5 and PDE9 are found in different intracellular compartments [86]; PDE5 primarily cleaves cGMP originated from the nitric oxide-soluble GC axis, while PDE9 cleaves cGMP formed via the natriuretic peptide–particulate GC axis [87]. PDE5 activity is regulated through phosphorylation and cGMP levels [85,88]. PDE5 is found in many types of tissues, but is highly expressed in pulmonary and penile vasculature, and at lower levels, is expressed in peripheral and coronary vasculature [89]. PDE5 activity is increased significantly in various experimental models of PH [90], leading to the accelerated degradation of cGMP and exacerbating adverse pulmonary vascular remodeling [90].

PDE5 inhibitors activate protein kinase G by accumulating cGMP [91]. PDE5 inhibition restores a normal cGMP transpulmonary gradient (increased arteriolar and capillary release) in patients with HF and high pulmonary vascular resistance [92]. PDE5 inhibitors successfully reduce pulmonary arterial and venous vascular tone and enhance penile vasodilation, and are usually used for the treatments of PAH and erectile dysfunction [93]. PDE5 inhibitors demonstrate a negative inotropic effect, suppressing the sensitivity of troponin I to Ca^2+^ through protein kinase G-dependent troponin I phosphorylation [94], and reducing intracellular Ca^2+^ concentration due to the protein kinase G-dependent phosphorylation of L-type Ca^2+^ channels (Figure 2) [27]. In patients with HFpEF, therapy with sildenafil was accompanied by a decrease in LV contractility compared with placebo [95]. The physiological role of PDE5 in the pulmonary vasculature has been extensively studied, whereas the role of PDE5 in the myocardium is less clear, mainly due to lower myocardial PDE5 expression [96].

In experimental studies with pressure overload, the inhibition of PDE5 was accompanied by reverse cellular and molecular remodeling, improvement in LV myocardial function, and the regression of LV hypertrophy [96,97,98,99]. PDE5 inhibition has been suggested as a promising treatment option for PH-HFpEF (see below for details).

## 8. Clinical Experience with PDE Inhibitors in PH-HFpEF

Evidence of overlap between late stages of left-sided PH and PAH served as the rationale for investigating the potential treatment options for PH-HFpEF (at least for Cpc-PH) that were previously limited to PAH. However, in patients with left-sided PH and pulmonary vascular disease, the isolated reversal of the precapillary component without a concomitant decrease in LV filling pressures (reversal of the postcapillary component) is dangerous, since the noncompliant LV could not overcome the acute increase in preload, and pulmonary edema may develop [100]. In addition, fluid retention induced by endothelin receptor antagonists and prostacyclin analogs is an undesirable side effect in left-sided HF [28]. Pulmonary vasodilators as endothelin antagonists and prostacyclin analogs being highly effective in PAH have demonstrated neutral or even negative effects in HF [101,102,103,104,105,106,107], including HFpEF [108,109]. The Macitentan in Pulmonary Hypertension due to Left Ventricular Dysfunction (MELODY-1) study showed no hemodynamic improvement after 12-week therapy with endothelin antagonist macitentan in patients with HF (76% with preserved EF) and invasively proven Cpc-PH. Macitentan-treated patients were quantitatively more likely to experience fluid retention versus placebo. No changes in NT-proBNP were observed [108]. The SERENADE study with macitentan in HFpEF and CpC-PH was discontinued due to poor enrollment; according to preliminary data, macitentan therapy did not result in a decrease in NT-proBNP or functional improvement compared with placebo (NCT03153111). In a prematurely completed pilot Endothelin Receptor Blockade in Heart Failure with Diastolic Dysfunction and Pulmonary Hypertension (BADDHY) study, 12 weeks of therapy with bosentan was not associated with clinical and hemodynamic benefits in patients with PH-HFpEF [109].

In contrast to other pulmonary vasodilators, evidence is accumulating that indicates PDE5 inhibitors may have a valuable role as an RV-unloading agent in left-sided PH [110]. PDE5 inhibition ameliorates LV diastolic stiffness via the protein kinase G-mediated phosphorylation of titin [111] and improves relaxation via the protein kinase G-mediated phosphorylation of phospholamban (accelerating Ca^2+^ resequestration into the sarcoplasmic reticulum) and troponin I (accelerating cross-bridge inactivation; Figure 2). PDE5 expression is exceptionally high in pulmonary vessels, and PDE5 inhibitors preferentially affect pulmonary rather than systemic circulation [75]. In patients with PH and HF with reduced EF, both acute [112,113,114] and chronic [115,116,117] oral PDE5 inhibition therapy was well tolerated and consistently decreased pulmonary artery systolic pressure and pulmonary vascular resistance without substantial changes in systemic blood pressure. No cases of pulmonary edema have been reported in these studies.

In HFpEF patients, PDE5 inhibitors have demonstrated contradictory results (Table 1). In the largest study to date, the Phosphodiesterase-5 Inhibition to Improve Clinical Status and Exercise Capacity in Heart Failure with Preserved Ejection Fraction (RELAX) trial evaluating the effect of sildenafil in HFpEF and involving 216 patients with HFpEF, therapy with sildenafil (20 mg TID for 12 weeks followed by 60 mg TID for 12 weeks) was not associated with significant improvement in exercise capacity or clinical status compared with placebo [118]. However, PH was not a mandatory inclusion criterion for this study, and the evaluation of pulmonary hemodynamics or RV function was beyond the scope of the study.

A more recent study by Hoendermis et al. enrolled 52 patients with PH associated with HFpEF [119], with PH confirmed by right heart chamber catheterization. The patients were randomized to receive sildenafil (20 mg TID with up-titration to 60 mg TID) or placebo for 12 weeks. Therapy with sildenafil was not accompanied by a decrease in pulmonary artery pressure or clinical improvement. However, only one third of patients had pulmonary vascular resistance > 3 Wood units, and the participants were characterized as patients predominantly with Ipc-PH.

Thus, these neutral results do not clarify whether patients with HFpEF and Cpc-PH may benefit from PDE5 inhibitors. This question has been tested in two small prospective studies. In the early single-center study, Guazzi et al. reported the positive effects of sildenafil therapy on hemodynamics and RV function in patients with HFpEF who predominantly met the hemodynamic criteria of Cpc-PH [120]. In a recent single-center study by Belyavskiy et al., 6-month sildenafil therapy (25 mg TID for 12 weeks followed by 50 mg TID for 12 weeks) in patients with HFpEF and Cpc-PH assessed by echocardiography was associated with an improvement in exercise capacity, pulmonary hemodynamic parameters, and RV function [121]. The major limitation of the study was the absence of invasive assessment of pulmonary hemodynamics, which is the reference method for the quantification of pulmonary artery pressure according to current guidelines [19]. Nevertheless, a pulmonary artery systolic pressure >50 mm Hg in HFpEF more likely indicates the concomitant pulmonary vascular disease rather than a consequence of left-sided HF [122]. The mean pulmonary artery systolic pressure was higher in the HFpEF group in the study by Belyavskiy et al. (57 mmHg) compared to both the RELAX study (41 mmHg) and the study by Hoendermis et al. (52 mmHg), but comparable to the study by Guazzi et al. (55 mm Hg). Both studies by Guazzi et al. [120] and Belyavskiy et al. [121] have demonstrated improvements in pulmonary hemodynamics in all patients treated with sildenafil.

In a meta-analysis of randomized trials comparing PDE5 inhibitors with placebo in chronic HF, the effects of PDE5 inhibition in patients with HFpEF were heterogeneous, with beneficial effects related to baseline pulmonary artery pressure levels and the extent of PDE5 inhibitor-mediated pulmonary artery pressure decrease [123].

A data from the Comparative, Prospective Registry of Newly Initiated Therapies for Pulmonary Hypertension (COMPERA), which included patients with “typical” PAH, “atypical” PAH (PAH and a high burden of cardiovascular comorbidities), and PH-HFpEF, showed an improvement in functional class, exercise capacity, and natriuretic peptides in 226 patients with PH-HFpEF who received pulmonary vasodilators, predominantly PDE5 inhibitors [124]. The patients with HFpEF initially had a very high transthoracic pressure gradient (mean 26 mm Hg) and pulmonary vascular resistance (mean 7 Wood units), assuming pulmonary vascular disease. The results of the study support the assumption that the Cpc-PH phenotype may benefit from therapies targeting pulmonary circulation. However, the effect of pulmonary vasodilators in patients with Cpc-PH and HFpEF was less pronounced compared to patients with “typical” PAH [124]. In a recent retrospective study by Kramer et al. the beneficial effects of PDE5 inhibitors on 6-min walk test distance, HF functional class, NT-proBNP levels, RV function, and hospitalization rates were demonstrated in 40 patients with HFpEF and Cpc-PH and precisely characterized hemodynamics [125]. However, important limitations of the study included the lack of a control group, and open-label therapy, which may result in a bias towards the overestimation of the treatment response.

Summary of studies with sildenafil in patients with HFpEF are summarized in Table 1.

Current data indicate that pulmonary vasculopathy may be at least partially reversible and thus may serve as a therapeutic target in HFpEF. Some of the aforementioned studies have shown that PDE5 inhibitor therapy was also associated with an improvement in RV function [120,121,125]. Biopsy studies have shown PDE5 expression was significantly increased in hypertrophied RV compared to the healthy RV myocardium [127,128], and has been associated with the severity of RV dysfunction [128]. The inhibition of PDE5 in a rat model of monocrotaline-induced PAH was accompanied by a significant increase in RV contractility in hypertrophied RVs compared to control animals [127]. PDE5 inhibition had almost no effect on the contractility of RV trabeculae extracted from nonfailing human hearts, but was accompanied by a moderate increase in the contractility of RV trabeculae from failing hearts [128]. Presumably, PDE5 expression is involved in the development of RV failure in left-sided PH, and PDE5 inhibition may independently contribute to the restoration of RV function in addition to the indirect positive effect on RV afterload. Normally, PDE5 inhibition leads to an increase in cGMP level and protein kinase G activity, which is accompanied by a decrease in intracellular Ca^2+^ concentration and should theoretically lead to a decrease in contractility. However, since protein kinase G activity in the hypertrophied myocardium is reduced [129], the increase in cGMP associated with PDE5 inhibition may not be sufficient for complete protein kinase G activation. Instead, the inhibition of cGMP-sensitive PDE3 occurs [130], which leads to an increase in cAMP and protein kinase A activity with a corresponding increase in intracellular Ca^2+^ levels and contractility [130]. These mechanisms may explain the increased contractility of a hypertrophied RV.

## 9. Clinical Experience with PDE Inhibitors in LV Diastolic Dysfunction

Apparently, the main target of PDE5 inhibition in HFpEF is the pulmonary vasculature, while myocardial effects are secondary, preventing an increase in LA pressure after the reactive pulmonary component is attenuated. PDE5 inhibitors may have a valuable role in the treatment of the chronic myocardial disorders associated with HFpEF, since one of the main pathogenetic mechanisms is a significant decrease in the activity of the nitric oxide–cGMP-protein kinase G signaling pathway [131]. According to the novel HFpEF paradigm, proinflammatory comorbidities, including metabolic disorders, hypertension, diabetes mellitus, and renal insufficiency trigger a low-grade systemic inflammation and coronary microvascular endothelial dysfunction with subsequent oxidative stress and the impairment in the nitric oxide–cGMP-protein kinase G signaling pathway. This leads to the deactivation of the main effector, protein kinase G enzyme, followed by cardiomyocyte hypertrophy, altered myofilament protein phosphorylation, and cardiac fibrosis [132]. The proinflammatory paradigm has generated a great interest in investigating the nitric oxide–cGMP-protein kinase G pathway in HFpEF. One of the mechanisms for activating this pathway is the inhibition of the PDEs-mediated degradation of cGMP (Figure 2). In addition to diastolic improvement (see above), PDE inhibition may have long-term structural effects such as antihypertrophic and antifibrotic effects [133]. However, the data on increased PDE5 expression in failing LV myocardium are contradictory [131,134,135,136,137].

In experimental studies, PDE5 inhibition beneficially influenced LV remodeling, suppressing pro-hypertrophic and profibrotic stimuli and attenuating cardiomyocyte stiffness [96,99,138], cardiac inflammation, and apoptosis [139]. There is also some clinical evidence of the cardioprotective effects of PDE5 inhibition in LV concentric remodeling and diastolic dysfunction. In a placebo-controlled study, 3-month sildenafil therapy was associated with an anti-remodeling effect (decrease in the LV mass-to-volume ratio), as well as a reduction in the inflammatory marker monocyte chemotactic protein-1 and fibrosis marker transforming growth factor-β, in 59 men with asymptomatic diabetic cardiomyopathy, preserved LV EF, and impaired LV deformation [140]. Therapy with sildenafil was accompanied by a decrease in proinflammatory chemokine CXCL10 in patients with diabetic cardiomyopathy [141]. Sildenafil also significantly decreased CXCL10 protein secretion and gene expression in human cardiomyocytes in vitro [141].

In the Sildenafil and Diastolic Dysfunction after Myocardial Infarction (SIDAMI) trial, 9-week sildenafil therapy in 70 patients with LV diastolic dysfunction after myocardial infarction was accompanied by an increase in cardiac index and decrease in systemic vascular resistance [126]. Sildenafil therapy was associated with a significant increase in LV end-diastolic volume, while no changes in resting pulmonary capillary wedge pressure and a tendency towards pulmonary capillary wedge pressure reduction during exercise were observed, indicating an increase in left heart chamber compliance with PDE5 inhibition. The PDE5 inhibitor tadalafil was shown to improve LV diastolic function in patients with systemic sclerosis-associated PAH when combined with endothelin receptor antagonist ambrisentan [142], as well as in patients with resistant arterial hypertension [143]. In patients with HF with reduced EF, PDE5 inhibition therapy with sildenafil [144] or udenafil [145] was accompanied by a decrease in LV filling pressure (early mitral inflow to mitral annulus relaxation velocities (E/e’) ratio) along with a reverse remodeling of LA volume index, indicating LV diastolic function improvement.

The previously mentioned studies evaluating the effect of PDE5 inhibitors on pulmonary hemodynamics in HFpEF also assessed the effect on diastolic dysfunction. In the RELAX trial, 24-week sildenafil therapy in patients with HFpEF was not accompanied by an improvement in exercise capacity, LV mass, and E/e’ ratio [118]. This study was performed when the predominant pathophysiological mechanisms or phenotype of HFpEF were not yet considered a major factor in determining the design of randomized trials in HFpEF. A large proportion of RELAX participants had comorbidities, including anemia and chronic obstructive pulmonary disease, which could explain the ineffectiveness of treatments aimed at increasing nitric oxide bioavailability. In the study of Liu L.C. et al., in 52 patients with HFpEF and predominantly Icp-PH, 12-week sildenafil therapy was accompanied by a significant reduction in the E/e′ ratio compared with baseline, but this treatment effect did not achieve a significant difference compared with the placebo group [146].

In the study of Guazzi et al., in patients with HFpEF and Cpc-PH, one-year sildenafil administration was accompanied by a significant decrease in LV filling pressures and an increase in LV end-diastolic diameter, suggesting an improvement in LV distensibility [120]. Similarly, in the study of Belyavskiy et al., therapy with sildenafil was associated with a decrease in LV mass and improvement in LV diastolic function in patients with HFpEF and Cpc-PH [121]. The reduction in the LV mass index was correlated with pulmonary capillary wedge pressure decrease during therapy, suggesting the role of other effects besides the lusitropic effects of sildenafil (antihypertrophic, antifibrotic) [147]. The patients in this study demonstrated pronounced LV hypertrophy (mean LV mass index was 133 g/m^2^) that was higher than in the RELAX trial and in the study of Liu et al. (<80 g/m^2^), which showed no benefits to LV diastolic function [118,146]. In animal models with pressure overload, PDE5 inhibition did not show antihypertrophic effects in mice with less severe pressure overload, whereas dramatic benefits were observed in mice with severe pressure overload, eccentric LV hypertrophy, and pulmonary congestion [96,99]. It is likely that the excessive LV remodeling associated with high PDE5 activation might preferentially benefit from PDE5 inhibition.

There is evidence of favorable vascular and pleiotropic effects of PDE5 inhibitors. In an ancillary sub-study of the RELAX trial, sildenafil therapy was associated with improved vascular function (decrease in arterial elastance and a tendency to increase the reactive hyperemia index), although there was a decrease in LV contractility [95]. These data are consistent with other studies, demonstrating a reduction in the total vascular resistance [92], an increase in aortic distensibility [148], and an improvement in endothelial function [149] in HFpEF patients treated with PDE5 inhibition. All of these factors are involved in the pathogenesis of HFpEF.

Since PDE5 inhibitors target cGMP generated by the nitric oxide-soluble GC axis, their pharmacological effects largely depend on the bioavailability of nitric oxide and, ultimately, on the activity of nitric oxide synthase [150]. Oxidative stress is evident in the pathophysiology of HFpEF [151], leading to excessive inhibition of nitric oxide synthase activity, a reduction of nitric oxide and cGMP bioavailability, and diminished substrate loading for PDE5. An alternative approach to increase cGMP-protein kinase G axis activity via PDE inhibition is to inhibit PDE9. PDE9 is responsible for the degradation of cGMP generated through the natriuretic peptide–particulate GC axis. Pronounced PDE9 activation has been found in the myocardium of patients with HFpEF [87], suggesting that low cGMP levels are associated with PDE9 overexpression and a beneficial effect of PDE9 inhibition is possible. In animal studies with pressure overload, the genetic or pharmacological blockade of PDE9 suppressed hypertrophy, fibrosis, LV dysfunction [87,152], and fibrosis [87].

It is suggested that PDE3 inhibition might also be a therapeutic option for patients with HFpEF. PDE3 is a dual-substrate phosphodiesterase with similar affinity for cAMP and cGMP. PDE3 inhibition might favorably influence key pathophysiological targets, including LV diastolic properties, peripheral circulation, chronotropic reserve, and pulmonary pressures. The infusion of the PDE3 inhibitor milrinone increased cardiac output reserves at lower LV filling pressures during exercise in patients with HFpEF [153]. Chronic therapy with oral PDE3 inhibitor cilostazol is now being explored in phase 2 trials in patients with HFpEF (ClinicalTrials.gov Identifier NCT05126836).

Completed and ongoing prospective clinical studies are summarized in Table 2.

One promising approach in the clinical application of PDE inhibition may be the prophylactic administration of these drugs in patients at high risk of HFpEF. In a recent experimental study, the PDE5 inhibitor vardenafil was administered early in the life of Zucker diabetic fatty (ZDF) rats (a model of metabolically induced HFpEF), before the development of diabetes. Compared with sildenafil, vardenafil is more specific to cGMP and interferes less with the cAMP–protein kinase A pathway. Vardenafil successfully reduced LV stiffness and improved relaxation, restored initially decreased cGMP levels and protein kinase G activity, and reduced nitro-oxidative stress, apoptosis, LV hypertrophy, and myocardial fibrosis [154].

In ZSF1 rats, derived from a cross between ZDF and a spontaneously hypertensive linages, sildenafil improved LV diastolic function when administered to 16-week-old rats for 4 weeks [155]. The control ZSF1 rats lacked myocardial fibrosis by 20 weeks of life, so the positive diastolic effect of PDE5 inhibition was predominantly due to the elimination of titin hypophosphorylation. However, the organization of clinical trials with PDE5 inhibitors as a preventive therapy in patients with HFpEF is currently a challenging task.

## 10. Conclusions and Perspectives

PH associated with HFpEF is the most common form of PH, and the morbidity and mortality related to PH-HFpEF continue to rise. PDE5 inhibitors, while approved for patients with PAH, have been evaluated with variable success as a therapy in HFpEF. Most clinical trials focusing on PDE inhibition for HFpEF involved heterogeneous populations, with many patients having mild/modest PH (Ipc-PH) without significant right heart dysfunction. Therapy targeting the precapillary component of PH in non-selective patients with HFpEF may have contributed to the relative failure of these studies. However, PDE5 inhibitors are emerging as promising approach for reversing pulmonary vascular disease and RV remodeling. It is hypothesized that the primary therapeutic effects of PDE5 inhibitors in HFpEF include pulmonary vasodilation and the restoration of RV contractility, so pronounced baseline dysfunction of both PH and RV dysfunction are essential for a clear clinical effect. Randomized trials are required to elucidate the actual value of PDE inhibition in selected patients with HFpEF and clearly verified pulmonary vascular disease and RV dysfunction. Patient selection with invasively confirmed Cpc-PH and acceptable endpoints (cardiovascular morbidity and mortality, changes in exercise capacity, pulmonary artery pressure, RV function, LV diastolic function, and N-terminal pro-brain natriuretic peptide) will be critical to evaluate the potential of this class of drugs. Another task of great importance is investigating the differentiation of structural pulmonary remodeling from functional vasoconstriction in patients with a pulmonary precapillary component, which could allow the prediction of the effectiveness of PDE5 inhibitors in HFpEF.

## Figures and Tables

**Figure 1 pharmaceuticals-15-01024-f001:**
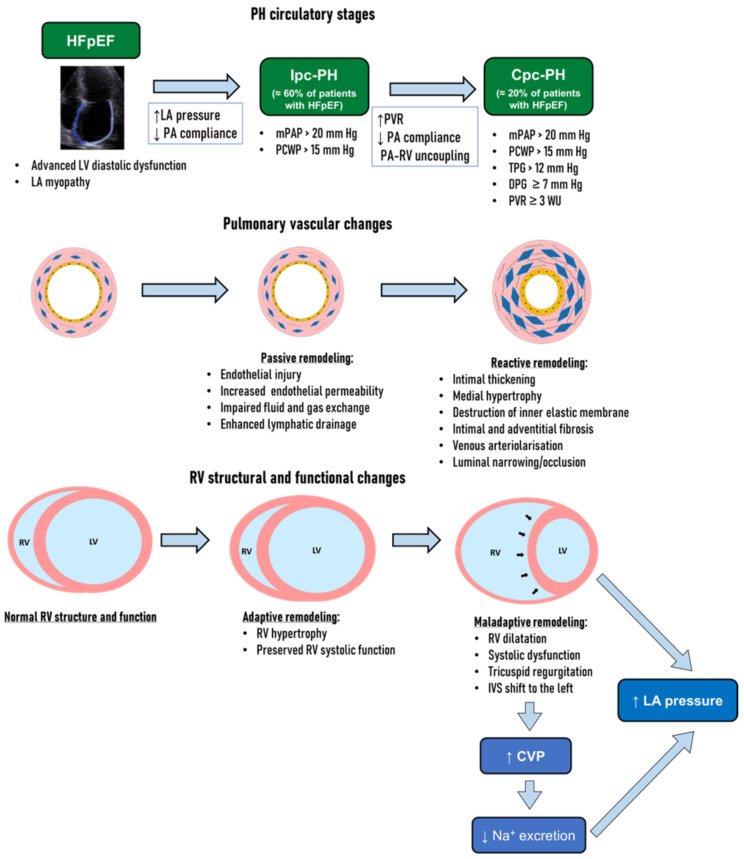
Stages of pulmonary hypertension in HFpEF. A chronic increase in mean left atrial (LA) pressure causes an increase in pulmonary artery (PA) pressure (pulmonary hypertension, PH), leading to passive remodeling in pulmonary venule and capillaries and isolated postcapillary PH (Ipc-PH). In approximately 20% of patients with HFpEF, the reactive “precapillary” alterations of pulmonary vasculature occur with the development of the combined pre- and post-capillary PH (Cpc-PH). Ipc-PH is associated with a decrease in pulmonary arterial capacitance (PAC) and mild adaptive changes of the right ventricle (RV). On the other hand, Cpc-PH leads to an increase in pulmonary vascular resistance (PVR), PA-RV uncoupling, and is associated with a marked maladaptive RV remodeling with RV systolic dysfunction and dilation, tricuspid regurgitation, and increase in central venous pressure (CVP). Increased CVP results in a reduction of sodium (Na^+^) excretion and fluid retention, and a further increase in LA pressure. With RV dilatation and CVP increase, shifting of the interventricular septum (IVS) to the left occurs, resulting in an impaired left ventricular filling. DPG indicates diastolic pulmonary gradient; mPAP, mean pulmonary artery pressure; PA, pulmonary artery, PCWP, pulmonary capillary wedge pressure; TPG, transpulmonary pressure gradient.

**Figure 2 pharmaceuticals-15-01024-f002:**
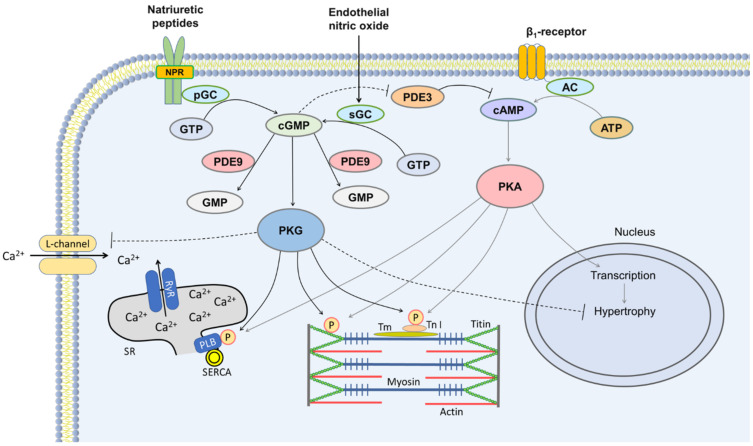
Cyclic nucleotide signaling in cardiomyocyte. Nitric oxide and natriuretic peptide receptor (NPR) activate soluble (sGC) and particulate guanylate cyclases (pGC), respectively, resulting in production of cyclic guanosine monophosphate (cGMP) and activation of protein kinase G (PKG). PKG phosphorylates numerous targets within myocyte. PKG-mediated phosphorylation (P) of phospholamban (PLB) activates sarcoplasmic–endoplasmic reticulum calcium ions (Ca^2+^)-ATPase pump (SERCA) and increases Ca^2+^ uptake into sarcoplasmic reticulum (SR); phosphorylation of troponin I (Tn I) reduces myofilament Ca^2+^ sensitivity increasing lusitropy. PKG-mediated phosphorylation of titin reduces cardiomyocyte stiffness, whereas PKG-mediated phosphorylation of L-type channels decreases Ca^2+^ influx, possessing a negative inotropic effect. Activation of β_1_-adrenergic receptors by epinephrine activates adenylate cyclase (AC), increasing the level of cyclic adenosine monophosphate (cAMP), which activates protein kinase A (PKA). High PKA activity leads to a positive inotropic effect due to phosphorylation of the L-type Ca^2+^ channel and the ryanodine receptor (not shown), increasing the systolic Ca^2+^ influx. PKA activation also possesses lusitropic effects through phosphorylation of the same targets as PKG-Tn I and PLB. PKA signaling mediates cardiac hypertrophy by increasing Ca^2+^ and calcineurin activation, as well as by increasing transcription. High cGMP and PKG levels promote negative inotropic effects and counteract PKA-mediated cardiac prohypertrophic signaling. cGMP also affects cAMP levels by inversely modulating PDE3. Phosphodiesterase cleaves cGMP, and PDE5 and PDE9 inhibitors increase cGMP levels. PDE5 primarily cleaves cGMP from the nitric oxide-rGC axis, while PDE9 cleaves cGMP from the natriuretic peptide-rGC axis. ATP indicates adenosine triphosphate; GTP, guanosine triphosphate; Tm, tropomyosin.

**Table 1 pharmaceuticals-15-01024-t001:** Summary of studies with sildenafil in patients with heart failure with preserved ejection fraction.

Study [References]	Study Population, *n*	Sildenafil Therapy	Mean PAP, mm Hg	RV Systolic Function	Pulmonary Vascular Resistance	Results
RELAX trial [118]	HFpEF (*n* = 216)	20 mg TID for 12 weeks, then 60 mg TID for 12 weeks	25	Normal	Normal	No clinical or hemodynamic benefit. Worsening renal function with sildenafil
Hoendermis E.S. et al. [119]	HFpEF + Ipc-PH (*n* = 52)	60 mg TID for 12 weeks	30	Mild dysfunction	Normal	No clinical or hemodynamic benefit
Guazzi M. et al. [120]	HFpEF + presumably Cpc-PH (*n* = 44)	50 mg TID for 52 weeks	40	Moderate-severe dysfunction	Increased	↓mPAP; ↓RV and LV filling pressures; ↑cardiac index; improved RV function, LV diastolic function, lung diffusion, and lung water
Belyavskiy E. et al. [121]	HFpEF + presumably Cpc-PH (*n* = 50)	25 mg TID for 3 months, then 50 mg TID for 3 months	≈40	Moderate dysfunction	Increased	↑6MWD; ↓PASP, RV and LV filling pressures, LVH; improved RV function, LV diastolic function, and NYHA functional class
Kramer T. et al. [125]	HFpEF + Cpc-PH, retrospective (*n* = 40)	20 mg TID for ≥12 months	46	Moderate dysfunction	Increased	↑6MWD; ↓NT-proBNP; improved RV function; ↓HF hospitalizations
SIDAMI trial [126]	HFpEF + post-MI (*n* = 70)	40 mg TID for 9 weeks	20	Normal	Normal	Improved CO and SVR; a trend to ↓PCWP at exercise

CO indicates cardiac output; Cpc-PH, combined pre- and post-capillary pulmonary hypertension; HF, heart failure; HFpEF, heart failure with preserved ejection fraction; Ipc-PH, isolated postcapillary pulmonary hypertension; LV, left ventricular; MI, myocardial infarction; NT-proBNP, N-terminal pro-brain natriuretic peptide; NYHA, New York Heart Association; PAP, pulmonary artery pressure; PCWP, pulmonary capillary wedge pressure; PH, pulmonary hypertension; PVR, pulmonary vascular resistance; RV, right ventricular; SVR, systemic vascular resistance; 6MWD, 6-min walk test distance; ↑, increase; ↓, decrease.

**Table 2 pharmaceuticals-15-01024-t002:** Summary of prospective clinical studies on LV diastolic effects of chronic therapy with PDE inhibitors.

Study [References]	Study Population	*n*	Study Design	Therapy	Duration	LV Diastolic Function
PDE5 inhibitors						
RELAX trial [118]	HFpEF	216	Multicenter, placebo-controlled	Sildenafil 20–60 mg TID	24 weeks	No changes
Guazzi M. et al. [120]	HFpEF + presumably Cpc-PH	44	Single-center, placebo-controlled	Sildenafil 50 mg TID	52 weeks	Improved
Belyavskiy E. et al. [121]	HFpEF + presumably Cpc-PH	50	Single-center, open-label	Sildenafil 25–50 mg TID	6 months	Improved
SIDAMI [126]	HFpEF + post-MI	70	Single-center, placebo-controlled	Sildenafil 40 mg TID	9 weeks	A trend to improvement
Sato T. et al. [142]	Systemic sclerosis-associated pulmonary arterial hypertension	21	Multicenter, open-label	Tadalafil + endothelin receptor antagonist ambrisentan	36 weeks	Improved
Santos R.C. et al. [143]	Resistant arterial hypertension	19	Single-center, placebo-controlled	Tadalafil 20 mg	2 weeks	Improved
Guazzi M., et al. [144]	HFrEF	45	Single-center, placebo-controlled	Sildenafil 50 mg TID	52 weeks	Improved
ULTIMATE-HFrEF [145]	HFrEF	41	Single-center, placebo-controlled	Udenafil 50–100 mg BID	12 weeks	Improved
Liu E.S. et al. [146]	HFpEF + Ipc-PH	52	Single-center, placebo-controlled	Sildenafil 60 mg TID	12 weeks	No changes
PDE3 inhibitors						
Cilostazol for HFpEF (ClinicalTrials.gov Identifier: NCT05126836)	HFpEF	25	Single-center, placebo-controlled	Cilostazol 100 mg BID	4 weeks	Ongoing Primary point: change in HF symptoms Secondary point: change in B-type natriuretic peptide

CMR indicates cardiac magnetic resonance; Cpc-PH, combined pre- and post-capillary pulmonary hypertension; HF, heart failure; HFpEF, heart failure with preserved ejection fraction; Ipc-PH, isolated postcapillary pulmonary hypertension; LV, left ventricular; LVH, left ventricular hypertrophy; MI, myocardial infarction.

## Data Availability

Data sharing not applicable.

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
