# Peer review of "Heart Failure with Preserved Ejection Fraction and Pulmonary Hypertension: Focus on Phosphodiesterase Inhibitors"

_pharmaceuticals, 2022, doi:10.3390/ph15081024_

Round 1
Reviewer 1 Report
This review gives a very good insight into the subject. The authors are to be congratulated.
This review has been written with great clarity from a clinical and pathophysiological point of view. The authors provide a balanced view. The high number of non-standard abbreviations is disturbing. 1. Please reduce the number of non-standard abbreviations. Non-standard abbreviations undermine the readbility of a manuscript.Combined pre- and post-capillary pulmonary hypertension is clear, Cpc-PH is simply unreadable. 2. Line 23: 'improving hemodynamics and RV function’. This is a very general and non-specific statement. 3. Line 32. This statement is unlikely to be true. It depends on the definition of HFpEF and more specifically on which cut-off value of ejection fraction is applied. 4. Wood unit (WU). Please explain this abbreviation at first use since non-clinicians will not be familiar with Wood units. 5. On several occasions, the letter type size is not correct (eg lines 182-184). 6. Reference 125: these data should be interpreted very cautiously. Please comment. 7. Conclusions and perspective. The authors are correct to point out that there is need for randomized trials. What would be an acceptable primary endpoint? Based on which criteria would they select patients? Patient selection will be critical to evaluate the potential of this class of drugs.
Reviewer 2 Report
The manuscript of “Heart failure with preserved ejection fraction and pulmonary hypertension: focus on phosphodiesterase inhibitors” by A. Ovchinnikov, A. Potekhina, E. Belyavskiy, and F. Ageev aims to review current knowledge about the pathogenesis and epidemiology of the combined pre- and post-capillary pulmonary hypertension (Cpc-PH) in heart failure with preserved left ventricular ejection fraction (HFpEF). The authors describe in detail the clinical experience of the use of phosphodiesterase inhibitors (mainly, the PDE5 inhibitor sildenafil) in PH-HFpEF and left ventricular diastolic dysfunction. The manuscript also includes the perspectives for the management of patients with PH-HFpEF.
The manuscript is written at a professional level and has significant scientific value.
Minor notes:
1. Lines 240-243: As is known, 3'5'-cyclic nucleotide phosphodiesterases are a family of phosphodiesterases. Nonetheless, there are many other families of phosphodiesterases, including phospholipases C and D, DNases and RNases, autotaxin, sphingomyelin phosphodiesterase, and restriction endonucleases, as well as numerous less-well-characterized small-molecule phosphodiesterases. Please, add the data to the text.
2. Line 256: Replace ‘RKA” with PKA and add a decoding of this abbreviation.
3. The manuscript could be supplemented with an illustrative diagram or a table summarizing the known phosphodiesterase inhibitors used in clinical practice in HFpEF and left ventricular diastolic dysfunction.
